# A Fluorescent Reporter Virus Toolkit for Interrogating Enterovirus Biology and Host Interactions

**DOI:** 10.3390/v17060796

**Published:** 2025-05-30

**Authors:** Mireya Martínez-Pérez, Sebastian Velandia-Álvarez, Cristina Vidal-Verdú, Beatriz Álvarez-Rodríguez, Ron Geller

**Affiliations:** Institute for Integrative Systems Biology (I^2^SysBio), Universitat de València—Consejo Superior de Investigaciones Científicas (UV-CSIC), 46980 Valencia, Spain

**Keywords:** enteroviruses, coxsackievirus B3, reporter assays, recombination, viral fitness, neutralizing antibodies, antivirals

## Abstract

Enteroviruses are a group of highly prevalent human pathogens responsible for a wide range of illnesses, ranging from common cold symptoms to life-threatening diseases. A deep understanding of enterovirus biology, evolution, and host interaction is required for the development of effective vaccines and antivirals. Recombinant reporter viruses encoding luminescent or fluorescent proteins have been developed to facilitate such investigation. In this work, using coxsackievirus B3 as our model, we analyze how the insertion of fluorescent reporter genes at three distinct sites within the viral polyprotein affects viral fitness, identifying the most tolerant site for reporter insertion. We then describe a set of experimental workflows for measuring viral fitness, sera neutralization, antiviral efficacy, and recombination using fluorescent reporter viruses. The high homology between different enteroviruses suggests these assays can be readily adapted to study additional members of this medically and economically relevant group of viruses.

## 1. Introduction

Enteroviruses are prevalent human and animal pathogens of high medical and economic importance. They include both newly emerging and ancient members, which have been causing disease for millennia. The symptoms and long-term consequences of enterovirus infections vary widely, ranging from mild respiratory and febrile illnesses to severe cardiac and neurological disorders [1,2]. Enteroviruses have also been implicated in the development of cancer and diabetes [3,4]. Owing to their medical importance and ease of manipulation, enteroviruses have played a key role in the history of virology [5]. They were the first viruses to be cultured and produced synthetically, and have led to the discovery of fundamental aspects underlying viral replication, host interaction, virus evolution, and vaccine development [5].

The enterovirus genome consists of a ~7.5 kb single-stranded RNA of positive polarity that is translated into a single polyprotein, which is subsequently cleaved in a highly regulated manner by two viral proteases, the 2A and 3C proteins, to generate the mature viral proteins [1,6,7]. The 2A protease only mediates a single cut in the polyprotein, the co-translational cleavage of the capsid precursor P1, but plays a key role in the modulation of the host cell environment [7,8,9]. All remaining cleavage events are mediated by the 3C protease, which liberates the non-structural precursor proteins P2 and P3 as well as all individual proteins in the P1–P3 precursors [6,7].

Early studies demonstrated that enteroviruses could accommodate insertions of up to ~15% of their genome length, provided that the foreign sequences were flanked by protease cleavage sites to enable their efficient liberation from the viral polyprotein [10]. However, size was found not to be the only determinant of insertion tolerability, as smaller inserts could impart larger fitness costs than longer ones [10]. Despite these constraints, numerous recombinant enteroviruses encoding luminescent or fluorescent proteins have been successfully constructed (e.g., [11,12,13,14,15]). Since their development, these have become fundamental research tools, facilitating real-time tracking of viral replication, translation, and production, as well as enabling high-throughput screening for antiviral compounds and antibody neutralization (e.g., [11,12,16,17]). Furthermore, they have been used to study viral recombination and single-cell virology, shedding light on the dynamics of viral evolutionary mechanisms and stochasticity in viral infection processes (e.g., [14,18,19]). It is important to highlight that the incorporation of reporters that become proteolytically separated from the viral polyprotein is useful for tracking overall viral replication and differs from alternative approaches in which reporter proteins are fused to individual viral proteins to examine protein localization (e.g., [15,16]).

To date, enterovirus reporter viruses have been engineered to express both small luminescent reporters (i.e., nanoluciferase) and fluorescent proteins, such as eGFP and mCherry [11,15,17,20]. Reporter proteins have been inserted at three primary locations within the viral polyprotein: immediately after the initiation methionine and at either the P1/P2 or P2/P3 junctions [11,17,20]. While viable reporter viruses have been successfully generated using all these strategies [11,17,20], a systematic comparison of how these insertion sites affect viral fitness has not been performed, to the best of our knowledge, and could help in the design of new enterovirus reporter constructs.

In general, luminescent reporters show higher sensitivity and larger dynamic ranges than fluorescent proteins [21]. However, viruses encoding fluorescent proteins offer several advantages. First, unlike luminescent reporters, they do not require the addition of exogenous substrates, significantly reducing costs and manipulation times. Second, they allow for live-cell, real-time tracking of infection, and are thus appropriate for assessing variability in infection processes. Finally, the large number of available fluorescent proteins makes fluorescent reporter viruses appropriate for use in multiplex assays. In this study, we first assess the impact of different insertion sites on the fitness of coxsackievirus B3 (CVB3) using the two commonly used fluorescent proteins, eGFP and mCherry. We then describe a series of experimental workflows to probe viral biology and host responses that capitalize on these tools in combination with live-cell microscopy, including the rapid titration of viral stocks, analysis of viral fitness via direct competition, screening of neutralizing sera and antivirals, and fluorescence-based recombination assays. Together, these approaches highlight the utility of enterovirus reporter systems for studying viral replication, host interactions, and evolutionary dynamics. The high conservation among the enteroviruses strongly supports the ability to generalize these assays to additional viruses, facilitating the study of different enteroviruses.

## 2. Methods

*Cell lines, viruses, and reagents.* HeLa-H1 (CRL-1958; RRID: CVCL_3334), HEK293T (CRL-3216), and BHK-21 ([C13] CCL10; RRID: CVCL_1914) cells were obtained from ATCC and routinely tested to confirm the absence of mycoplasma. The cells were maintained in Dulbecco’s modified Eagle’s medium (DMEM) containing Pen-Strep and L-glutamine, supplemented with either 10% heat-inactivated fetal bovine serum for standard culturing or 2% for infection experiments, at 37 °C with 5% CO_2_. The anti-coxsackievirus B3 monoclonal antibody was obtained from Merck (clone 280-5F-4E-5E, MAB948). Interferon-Iβ (recombinant human Interferon beta protein ab71475, Abcam, Cambridge, UK), ribavirin (Sigma-Aldrich R9644), Guanidine hydrochloride (Sigma-Aldrich, St. Louis, MO, USA), resazurin sodium salt (Sigma-Aldrich R7017), and Agar (BP1423-2, Fisher BioReagents™, Pittsburgh, PA, USA) were also obtained.

*Fluorescent reporter virus construction.* The CVB3 Nancy lab strain and the infectious clone encoding an unmodified mCherry (mCh) reporter (CVB3 P2:mCh) have been previously described [20]. The remaining CVB3 plasmids (GFP:P1, P1:GFP, P2:GFP, mCherry:P1, P1:mCh, and P1:nLuc) were generated by amplifying the CVB3 genome from either the CVB3 Nancy or the CVB3 P2:mCh plasmid, along with the corresponding fluorescent protein sequence from either P2:mCh, the pEGFP-N1 plasmid (Clonetech, San Jose, CA, USA), or the nanoluciferase vector pNL1.1 (Promega, Madison, WI, USA). PCR amplification was performed using Q5^®^ High-Fidelity 2X Master Mix (M0492) with primers designed for HiFi recombination (NEBuilder HiFi DNA Assembly Master Mix, E2621, Ipswich, MA, USA), and assembled products were transformed into E. coli NZY5α competent cells (MB00402, NZY Tech, Lisbon, Portugal) via heat shock transformation. Mutant viruses were generated by site-directed mutagenesis of the CVB3 P2:mCherry plasmid. For each mutant, a pair of non-overlapping primers—one carrying the desired mutation—was used in a PCR reaction with Q5 polymerase. The PCR product was then treated with DpnI, phosphorylated, and ligated (FastDigest DpnI FD1704, PNK EK0031, T4 DNA ligase EL0014, Thermo Scientific, Waltham, MA, USA). The ligation product was transformed into chemically competent E. coli NZY5α cells, and successful mutagenesis was verified by Sanger sequencing. All plasmid sequences are available in Appendix A.

*Recombinant virus recovery and titration.* Purified plasmids were linearized using SalI enzyme (Thermo Scientific, FD0644) when the fluorescent protein was located between P2 and P3 and with MluI (Thermo Scientific, #FD0564) for the remaining constructs to minimize extra nucleotides after the poly(A) tail. To recover passage 0 virus, linearized plasmids were transfected into HEK293T cells using Lipofectamine 2000 (ThermoFisher Scientific, 11668019) along with a human codon-optimized T7 polymerase plasmid (#65974, Addgene, Watertown, NY, USA), following the manufacturer’s instructions. A ratio of 2.5 µL of Lipofectamine per µg of total plasmid DNA was used; specifically, 1.2 µg of plasmid DNA was transfected with 3 µL of Lipofectamine in a 12-well plate. At around 20 hpt, cells were freeze-thawed three times, the supernatant was clarified by centrifugation, and viral stocks were maintained at −80 °C. Viral titers were quantified by infecting HeLa-H1 cells with serial dilutions of each virus in 96-well plates, followed by quantification of fluorescent cells at 8 hpi using the Incucyte SX5 Live-Cell Analysis System (Sartorius, Göttingen, Germany). To generate the passage 1 viral stocks, HeLa-H1 cells were infected at a multiplicity of infection (MOI) of 5 (n = 3), and viruses were harvested at around 20 hpi. As for passage 0, viral titers were quantified by fluorescence-based assays and additionally confirmed by plaque assays. To perform the plaque assays, HeLa-H1 cells were infected with 10-fold serial dilutions of the corresponding CVB3 variant (eGFP:P1, P1:eGFP, or P1:mCherry) for 45 min at 37 °C in a final volume of 200 µL. After infection, cells were overlaid with 3 mL of 0.8% agar in DMEM supplemented with 2% FBS and incubated at 37 °C for 2 to 3 days. Cells were then fixed with 10% formaldehyde, the agar overlay was removed, and plaques were visualized by staining with crystal violet.

*Competition assays.* HeLa-H1 cells were co-infected with 500 FCUs of one of the test viruses (eGFP:P1, P1:eGFP, mCherry:P1, P1:mCherry, or P2:mCherry) and 500 FCUs of a control virus (P1:mCherry for eGFP reporter viruses or P1:eGFP for mCherry reporter viruses) for 45 min in a 12-well plate. When 6-well plates were used, infection was carried out with 1000 FCUs per virus. Subsequently, viruses were removed and media containing 2% FBS was added. Following 8 hpi and 16 hpi, red and green fluorescent cells were counted using the Incucyte SX5. Relative fitness was calculated by dividing the 16 hpi to 8 hpi infection ratio of the test virus by the corresponding ratio for the control virus.

*Antibody neutralization assays.* To evaluate the sensitivity of CVB3 WT and mutant viruses to antiviral treatments, 2.5 × 10^3^ fluorescent virus units of WT or mutant CVB3 mCherry virus were mixed in duplicate with serial dilutions of antibody in DMEM containing 2% FBS and incubated for 1 h at 37 °C. Following incubation, the virus–antibody mixtures were added to monolayers of HeLa-H1 cells seeded in 96-well plates and incubated at 37 °C with 5% CO_2_ for 8 h. The IC50 values were then obtained by subtracting the number of mCherry-positive cells in the mock-infected conditions from all virus-infected wells, followed by standardization to the average number of mCherry-positive cells in the infected, mock-antibody treated wells, and fitting a 3-parameter log-logistic function using custom R scripts (version 4.4.3) and the drc package (3.0–1). All analysis scripts are available online (https://github.com/RGellerLab/EC-calculation-scripts, accessed 16 April 2025).

*Antiviral drug screening.* For interferon-Iβ (IFN-Iβ) and ribavirin sensitivity assays, a similar protocol to the antibody neutralization assay was performed, with the exception that cells were pretreated for 24 h before infection with increasing concentrations of IFN-Iβ (final concentrations: 0.04–100 IU/mL). At 16 hpi, fluorescence intensity was measured using the Incucyte SX5 microscope to quantify the number of virus-infected cells. To evaluate potential cytotoxicity, cell viability was assessed in parallel using the resazurin reduction assay (Alamar blue assay) as previously described [22]. All experiments were performed in at least three independent replicates.

*Analysis of RNA replication and translation kinetics.* The CVB3 reporter virus encoding nanoLuciferase (nLuc) between P1 and P2 was constructed by replacing the eGFP sequence in P1:GFP to generate the plasmid P1:nLuc. P1:nLuc and P2:mCh plasmids were linearized with restriction enzymes as indicated above and vRNA was produced by in vitro transcription using the T7 high-yield Transcriptaid (Thermo K0441) as previously described [17]. After in vitro transcription, RNA samples were purified by precipitation with 3.75M LiCl. For the P2:mCh assay, approximately 4.5 × 10^5^ HeLa H1 cells per well were seeded in a 12-well plate on the day of transfection using DMEM supplemented with 10% FBS. Twelve hours later, transfections were performed in triplicate for each condition using 1.2 μg of RNA and 3 μL of Lipofectamine 2000 per well. Transfection of the P1:nLuc RNA was performed using a 600 ng RNA to 1.5 μL Lipofectamine 2000 ratio per well in 48-well plates containing approximately 9 × 10^4^ cells per well. One hour post-transfection, cells were washed with PBS, and the media were replaced with 2% FBS DMEM either supplemented or not with 0.5 mM GuHCl. The fluorescence of mCherry and nLuc activity was monitored using an Incucyte SX5 live-cell microscope and the Nano-Glo Luciferase Assay System (Promega N1120), respectively, at each time point. Results were expressed as mCherry-positive cells per well or as relative nano-luciferase luminescence. All conditions and time points were performed in triplicate.

*RNA recombination assays.* A CVB3 replication-incompetent plasmid encoding eGFP was engineered from the infectious clone and GFP:P1 reporter plasmid. First, the infectious clone was amplified by PCR with Phusion High-Fidelity DNA Polymerase (Thermo Scientific) removing the catalytic residues (^2051^Gly-Asp-Asp^2053^) from the viral polymerase [23], followed by phosphorylation with PNK and ligation with T4 DNA ligase. Then, eGFP from the GFP:P1 construct was inserted using XhoI/BstBI digestion and ligation, generating GFP:P1^ΔGDD^. A CVB3 subgenomic replicon, lacking the capsid-coding sequence and encoding the mCherry (ΔP1:mCh), was generated via PCR from GFP:P1 and P2:mCh plasmids and assembled by seamless cloning. Viral RNAs from GFP:P1^ΔGDD^ and ΔP1:mCh plasmids were produced by in vitro T7 transcription as previously described [17]. Equimolar amounts of each viral RNA (600 ng each; total 1.2 µg) or 1.2 µg of either individual RNA were transfected in triplicate into BHK-21 cells that were plated in 12-well dishes 24 h prior to transfection (~2 × 10^5^ cells/well). Transfections were performed using 3 µL of Lipofectamine 2000 per well and following the manufacturer’s protocol. Four hours after transfection, cells were washed with PBS, and DMEM 2% FBS was added. At 18–20 hpt, viruses were harvested after three freeze–thaw cycles, clarified by centrifugation (2000 *g*, 10 min), and titrated on HeLa-H1 cells in triplicate. Phase contrast, red, and green fluorescence were monitored at 10 hpt as well as at 8 and 20 hpi on an Incucyte SX5 live-cell microscope. Red and green fluorescent cells were quantified following transfection and infection, and results were expressed as fluorescent cells per well and as fluorescent count units per milliliter (FCU/mL), respectively.

## 3. Results and Discussion

### 3.1. Insertion Site and Reporter Sequence Determine CVB3 Viability and Fitness

To better understand how insertion position in the polyprotein and the reporter sequence affect viral fitness, we created a set of recombinant CVB3 infectious clones encoding either eGFP (GFP) or mCherry (mCh) at the main sites within the viral open reading frame previously described as permissive to reporter gene insertion (Figure 1A). First, the reporters were inserted downstream of the initiating methionine (GFP:P1 or mCh:P1) and were followed by a five amino acid sequence from the C-terminus of 2C to reconstitute the required 3C cut site (Q/G). Upon translation, the 3C protease should release the fluorescent protein from the polypeptide, unmasking the natural myristylation sequence of the viral capsid protein [24]. The second set of constructs was generated by inserting the fluorescent proteins at the junction of P1 and P2 (P1:GFP or P1:mCh) flanked by the first and last five amino acids of the 2A and VP1 proteins at the N- and C- termini of the reporters, respectively, to reconstitute the natural 2A protease cleavage site (T/G). Finally, the third set of constructs was created by inserting the reporter genes at the junction of the P2 and P3 regions (P2:GFP or P2:mCh), adding the ≥5 amino acids of the beginning or end of the 3A and 2C proteins at the N- and C- termini of the reporter, respectively, to reconstitute the 3C protease cleavage site (Q/G).

We next generated viral stocks by transfection of the virus reporter constructs into HEK293T cells (see Methods) and quantified virus production (passage 0) in HeLa-H1 cells by counting the number of fluorescent cells after a single replication cycle (8 h post-infection, hpi; Figure 1B and Appendix A). All constructs yielded high levels of virus production [range: 1.57 × 10^7^–2.7 × 10^8^ fluorescent cell units (FCU)/mL], except for P2:GFP, where no GFP-positive cells were observed. To assess whether the lack of fluorescence was due to loss of the eGFP insert, we titrated P2:GFP passage 0 by plaque assay. Visible plaques could be observed, indicating the presence of infectious CVB3 (Appendix A). Moreover, a complete loss of the eGFP insertion was confirmed by PCR in passage 1 viral population (Appendix A). These findings reveal the high cost of the eGFP sequence when inserted between the P2 and P3 precursor regions, which is in stark contrast to the results obtained with the mCherry reporter virus (P2:mCh), highlighting how reporter sequence can affect viral fitness.

The ability to quantify virus production based on fluorescence signal within a few hours (e.g., 6–8 h) using small-well formats and no additional reagents (e.g., agar, crystal violet) offers significant time and reagent savings compared to traditional plaque or limiting dilution assays. To assess the reliability of fluorescence-based quantification, we compared the titers obtained using reporter proteins with those determined by a plaque assay for three of the reporter viruses (GFP:P1, P1:GFP, and P1:mCh; see Figure 1C). The plaque assay yielded higher titers for P1:GFP and P1:mCh (1.0 × 10^7^ ± 1.8 × 10^6^ FCU/mL versus 3.4 × 10^7^ ± 4.5 × 10^6^ PFU/mL and 2.7 × 10^8^ ± 2.2 × 10^7^ FCU/mL versus 1.0 × 10^9^ ± 3.2 × 10^8^ PFU/mL for P1:GFP and P1:mCh, respectively) but lower titers for GFP:P1 (1.6 × 10^7^ ± 2.8 × 10^6^ FCU/mL versus 8.0 × 10^6^ ± 5.0 × 10^5^ PFU/mL) as compared to the fluorescence-based titration method (*p* < 0.05, 0.001, and 0.05, respectively, using a two-tailed *t*-test on log-transformed data). Despite statistically significant differences being observed between the two methods, the overall order of magnitude remained consistent across all three tested variants, indicating that fluorescence-based titration provides an accurate, low-cost, and rapid alternative for viral quantification.

To evaluate the relative fitness of the fluorescent reporter viruses, we compared viral production from a single infection cycle in HeLa-H1 cells (Figure 2A). Virus production was highest for P1:mCh and P2:mCh reporter viruses (1.97 × 10^7^ ± 5.69 × 10^5^ and 1.97 × 10^7^ ± 1.76 × 10^6^ FCU/mL for P1:mCh and P2:mCh, respectively), with no significant difference between them (*p* > 0.05 by one-way ANOVA followed by Tukey’s post hoc test). Both eGFP reporter viruses showed a similar, intermediate level of virus production (8.90 × 10^6^ ± 3.82 × 10^5^ and 7.70 × 10^6^ ± 1.49 × 10^6^ FCU/mL for GFP:P1 and P1:GFP, respectively), which was nonetheless significantly higher than that of the mCh:P1 reporter virus (5.39 × 10^6^ ± 8.38 × 10^5^ FCU/mL; Figure 2A). Hence, viral fitness was influenced by the interaction of both the fluorescent protein sequence and the insertion site. Supporting these results, the two eGFP reporter viruses formed markedly smaller plaques than the P1:mCh reporter virus (Figure 2B). The differences underlying these observations remain to be defined, but could potentially stem from differences in codon usage, RNA structure, or accessibility of the viral protease. As both reporter genes are of similar size (714 versus 705 bp, for eGFP and mCherry, respectively), sequence length is unlikely to play a role. Collectively, our findings suggest that inserting a reporter protein between P1 and P2 is the most permissive strategy, allowing robust viral replication across different fluorescent proteins. This is consistent with previous studies showing the N terminus of 2A to be the most tolerant to mutations and indels in both CVB3 and enterovirus A71 [25,26]. Indeed, viable reporter viruses with insertions at this position have been successfully generated in various enterovirus species, including enterovirus A71, poliovirus, and CVA5 [11,12,15,18]). Notably, reporter viruses encoding a 2A protease cleavage sequence rather than that of the 3C protease to liberate reporter proteins preceding the viral ORF have been described (e.g., [27]) and could potentially improve the fitness of reporter viruses such as GFP:P1 or mCh:P1.

### 3.2. Fluorescence-Based Competition Assays Show Increased Resolution Compared to Virus Production Assays

Direct competition between a test virus and a reference virus under identical conditions and over multiple replication cycles can provide a quantitative measure of viral fitness with improved resolution compared to virus production assays. Multiple methods have been used to quantify the relative abundance of the test virus and reference virus, including Sanger sequencing, qRT-PCR using virus-specific primers, or the measurement of fluorescent signal (e.g., [17,20,21,28,29,30]). Of these, fluorescence-based competition assays offer several advantages, namely bypassing the need for RNA extraction or reverse transcription, avoiding biases arising from differences in particle-to-infectious particle ratios among viral mutants, and not requiring additional reagents. To better define the fitness of the reporter viruses, the eGFP-expressing viruses were individually competed with the P1:mCh reporter virus as the reference, and the number of eGFP and mCherry positive cells were quantified at both 8 hpi (reflecting primary infection) and 16 hpi (reflecting secondary infections; Figure 2C). The fitness of each eGFP-expressing virus was determined by comparing the relative number of infected cells resulting from viral spread (16 hpi) to the initial infection (8 hpi), normalized to the corresponding ratio for the common reference virus (P1:mCh). In agreement with the results of the virus production assay, the fitnesses of the eGFP reporter viruses were significantly lower than that of the reference virus (average fitness of 0.248 ± 0.039 and 0.207 ± 0.011 for GFP:P1 and P1:GFP, respectively) but did not differ significantly between them (*p* > 0.05 by two-tailed *t*-test on log-transformed data; Figure 2A,D). Similarly, the mCherry-expressing viruses were competed with the P1:GFP reporter virus as a reference. As with the virus production assay, mCh:P1 showed the lowest fitness of the three mCherry-encoding reporter constructs (avg. relative fitness of 1.462 ± 0.081; *p* < 0.01 versus both P1:mCh and P2:mCh by two-tailed *t*-test on log-transformed data; Figure 2E). However, significant fitness differences were observed between the P1:mCh and P2:mCh reporter viruses (avg. relative fitness of 4.650 ± 0.284 and 3.752 ± 0.345 for P1:mCh and P2:mCh, respectively; *p* < 0.05 by two-tailed *t*-test; Figure 2E), which were not evident in the viral production assay (Figure 2A). Hence, the fluorescence-based competition assay offers a sensitive and simple means of assessing viral fitness with greater resolution than that observed with virus production assays.

### 3.3. High-Throughput Screening of Antibody Neutralization

Fluorescent reporter viruses provide a quantitative, real-time measure of infectivity without the need for additional reagents or manipulation, making them ideal for rapid, high-throughput quantification of sera neutralization capacity and antiviral effects. We have previously developed a method for screening the neutralizing effects of human sera and monoclonal antibodies (mAb) using fluorescent CVB3 reporter viruses [17]. This method is based on pretreatment of fixed amounts of fluorescent virus with serial dilution of the sera, followed by infection of cells in a 96-well format and quantification of the number of viral infected cells (fluorescent protein-expressing cells) following a single cycle of infection (8hpi; Figure 3A). To determine the dose resulting in 50% neutralization (IC50), background fluorescence from uninfected wells (cells-only control) is first subtracted. The data are then normalized to the average fluorescence in mock-treated, virus-infected cells (virus-only control), and a log-logistic function is fitted (Figure 3A). To facilitate the analysis of numerous sera samples, we have developed a custom analysis pipeline in R that automates data handling, log-logistic curve fitting, and graphing for visual inspection of data fitting (available at https://github.com/RGellerLab/EC-calculation-scripts, accessed 16 April 2025). As an example, this assay was used to determine the IC50 of a commercially available mAb (clone 280-5F-4E-5E, MAB948) against the WT virus, a known escape mutant (K227S), and two mutants that escape neutralization by antibodies targeting alternative capsid regions (K650Y and K723E; Figure 3B; [17,31]). The mAb escape mutant K227S exhibited complete resistance to neutralization over the range of concentrations utilized in the assay, compared to the WT virus or non-relevant escape mutants, which were clearly neutralized (reciprocal IC50 of 4670 ± 788, 2996 ± 266, and 2863 ± 245 for the WT, K650Y, K723E viruses, respectively). The ability to quantify serum neutralization of viral infection without the need for additional reagents and within a few hours makes fluorescence-based neutralization assays a low-cost, high-throughput alternative to the traditional plaque reduction neutralization test (PRNT) or Enzyme-Linked Immunosorbent Assay (ELISA). An additional advantage of this assay lies in its potential for multiplexing several enteroviruses, each encoding a different fluorescent reporter, to further reduce hands-on time and costs.

### 3.4. Analysis of Drug–Response Curves and Mechanism of Action Studies Using Fluorescent Reporter Viruses

The ability to screen large compound libraries with minimal manipulation and in a small-well format is essential for high-throughput screening of antiviral compound libraries. The antibody neutralization assay protocol and the analysis scripts described above can also be applied to antiviral drug screening. As an example, we analyzed the antiviral effect of interferon-Iβ (IFN-Iβ) and the RNA mutagen ribavirin [32] against CVB3 (Figure 4). IFN-Iβ pretreatment of cells results in a dose-dependent reduction in viral infection, with an IC50 of 5.03 ± 0.97 international units (IU)/mL (Figure 4A). Similarly, ribavirin inhibited the replication of both eGFP- and mCherry-expressing reporter viruses (IC50 of 48.57 ± 5.0 µM and 73.28 ± 4.76 µM for P1:GFP and P1:mCh, respectively; Figure 4B). The reduced level of virus infection was not due to effects on cell health, as cell viability was not significantly affected even at the highest doses (Figure 4B, red line). As for the sera neutralization assays, these antiviral assays can be adapted to multiplex format using reporter viruses encoding different fluorescent proteins. However, additional optimization to account for viral replication cycle kinetics must be performed depending on the viruses utilized.

Following the identification of antiviral compounds, a common downstream analysis is the interrogation of specific effects on viral entry, translation, and replication to define the step at which the antiviral acts (see [33] as an example). The ability to skip viral entry by direct electroporation or transfection of viral RNA (vRNA) allows for direct assessment of whether entry is affected. Similarly, RNA replication and translation kinetics can be monitored by examining reporter expression over time following vRNA transfection in the absence or presence of an enterovirus RNA replication inhibitor (Guanidine Hydrochloride, GuHCl). Traditionally, these assays have frequently relied on the transfection of full or subgenomic (replicon) vRNA encoding a luminescent reporter (e.g., [34,35]). While luminescent reporters show high sensitivity and a large linear range, the need to collect individual time points and the high cost of the substrates remain important limitations. To assess whether fluorescent reporters can be used in place of luminescent ones, we compared the replication kinetics of reporter viruses encoding either nanoluciferase (P1:nLuc) or mCherry (P2:mCh) following vRNA transfection into HeLa-H1 cells at 2 h intervals in the absence or presence of GuHCl to monitor viral replication or translation, respectively (Figure 4C,D). The nLuc reporter showed log-linear exponential growth between 4 and 8 h post-transfection (hpt; Figure 4C), while the fluorescent reporter displayed a sigmoidal curve (Figure 4D), likely reflecting the lower linear range of fluorescent reporters compared to luminescent ones. Nevertheless, replication could clearly be distinguished from translation in both cases, with the fluorescent reporter showing significant differences at early time points following transfection (2 hpt; *p* > 0.05 and *p* < 0.01 for P1:nLuc versus P2:mCh, respectively, using a two-tailed *t*-test on log-transformed values). Hence, although the linear range of the fluorescent reporter is lower than that of the luminescent one, the reduced cost and increased ease of use support the utilization of fluorescent viruses to study viral entry, translation, and replication kinetics.

### 3.5. Tracking RNA Recombination Using Fluorescent Reporter Viruses

Enteroviruses display high rates of recombination, which helps purge deleterious mutations and increase viral adaptability [18,36]. Consequently, understanding the mechanisms underlying recombination in enteroviruses is of significant interest. Cell-based recombination assays have been developed employing two viral RNAs (vRNAs) that do not produce infectious particles on their own but can generate fully infectious virus particles by recombination [23,36,37,38,39,40,41]. Briefly, these assays utilize a replication-competent donor vRNA lacking the capsid region and an acceptor vRNA that is rendered replication-deficient by either mutation of the *cis*-acting replication element (CRE; located in the 2C coding sequence for many enteroviruses) or inactivation of the viral polymerase (e.g., via deletion of the catalytic triad, ΔGDD) [39,41]. Since neither donor nor acceptor vRNAs can yield infectious particles on their own, quantification of infectious virus production can be used to measure the rate of recombination for different mutants or cellular conditions. Building upon this assay, we designed a fluorescence-based recombination assay in CVB3 using two distinct fluorescent reporters in the donor and acceptor vRNAs to easily track recombination (Figure 5). Specifically, a subgenomic replicon encoding the mCherry sequence in place of the P1 capsid region was constructed for the donor vRNA (Figure 5A, donor), and the GFP:P1 reporter was rendered replication-incompetent by deletion of the catalytic triad (ΔGDD) in the 3D active site to generate the acceptor vRNA (Figure 5A, acceptor). Individually, the introduction of the donor vRNA into cells should result in mCherry expression while minimal or no eGFP fluorescent signal should be observed from the acceptor RNA due to its inability to replicate. Recombination events between the donor and acceptor vRNA occurring between the capsid region and the ΔGDD deletion can yield replication-competent vRNAs encoding both eGFP and the capsid proteins, resulting in both eGFP signal and the production of infectious particles (Figure 5A, recombinant). These can then be used to quantify both the number of cells in which recombination has occurred as well as the number of recombinant viruses produced.

To test this system, we transfected either the donor vRNA, the acceptor vRNA, or both in equimolar amounts into BHK-21 cells (Figure 5B). We chose these cells as they are not permissive to CVB3 infection due to low receptor expression [22], facilitating the quantification of recombination events without the need to control for subsequent reinfection events. As expected, after individual transfection with the donor or acceptor vRNA separately, mCherry expression was detected but not eGFP (Figure 5B). In contrast, when vRNA from both the donor and acceptor were co-transfected into cells, eGFP expression could be observed in a subset of cells (Figure 5B, donor + acceptor), suggesting recombination had occurred. As expected for rare events such as recombination, a significantly lower number of eGFP-positive cells (196 ± 17.8 cells per well) was observed compared to mCherry-positive cells (40.703 ± 1.779; *p* < 10^−5^ by *t*-test on log-transformed values) despite transfection with equal amounts of donor and acceptor vRNAs (Figure 5B,D). To validate the occurrence of recombination events, supernatant from transfected wells was collected at 20 hpt, and viral titers were quantified by infection of HeLa-H1 cells. As expected, no viral particles were detected in the supernatants from individual transfections with the donor or acceptor vRNAs (Figure 5C). In contrast, eGFP-positive cells were readily detected at 8 hpi following infection with supernatants obtained from co-transfected cells (22.4 ± 18.3 FCU/mL; Figure 5C,E). A large increase in the number of eGFP-positive cells was observed at 20 hpi (mean increase of 14.6 ± 10.7-fold increase), indicating initial eGFP signal was derived from replication-competent recombinant viruses (Figure 5C,E). Interestingly, while mCherry-positive cells were observed at 8 hpi (113 ± 48.6 FCU/mL), only a small increase was observed at 20 hpi (1.63 ± 0.32-fold increase). This observation indicates that mCherry-positive cells result from infection of donor vRNA that is trans-encapsidated in cells where recombination had occurred and are thus only capable of a single round of replication. In sum, the developed recombination assay employing donor and acceptor vRNAs harboring distinct fluorescent reporter proteins can be used to quantify both the number of cells in which recombination occurs as well as the number of recombinant viruses produced, in an easy and scalable manner.

In summary, we show that fluorescent reporter viruses can be utilized to facilitate the study of enterovirus biology and host interactions, reducing the time and cost of both traditional and luminescence-based assays. While we highlight multiple applications of such viruses, additional assays can be readily envisioned to extend their utility, such as analysis of cell line susceptibility, quantification of replication cycle dynamics, or non-invasive imaging of replication kinetics *in vivo* using appropriate fluorescent reporters (e.g., infrared fluorescent proteins (iRFPs); [41]).

## Figures and Tables

**Figure 1 viruses-17-00796-f001:**
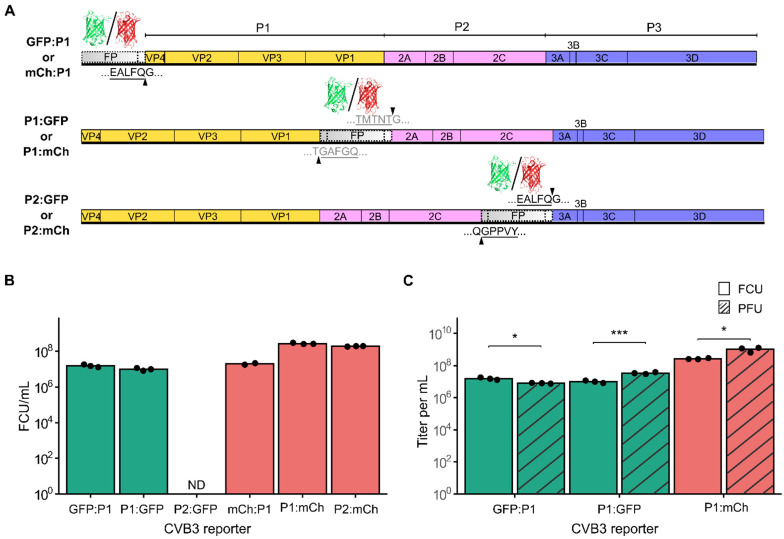
Generation and characterization of CVB3 reporter viruses. (**A**) Schematic representation of the reporter virus constructs. eGFP or mCherry was inserted at three different sites across the CVB3 proteome: upstream of the P1 precursor (mCh:P1 and GFP:P1), between the P1 and P2 precursors (P1:GFP and P1:mCh), and between the P2 and P3 precursors (P2:GFP and P2:mCh). The additional amino acid sequences inserted to generate the 3C (black) or 2A (gray) cleavage sites are underlined. (**B**) Titration of passage 0 CVB3 variants in HeLa-H1 cells using a fluorescence-based method. Bars represent the mean of ≥2 technical replicates, shown as black dots. ND, not detected. (**C**) Comparison of titers obtained by the fluorescence-based or plaque-based assays for three CVB3 reporters (n = 3). Statistical significance was assessed using paired *t*-tests on log-transformed data. * *p* < 0.05, *** *p* < 0.001.

**Figure 2 viruses-17-00796-f002:**
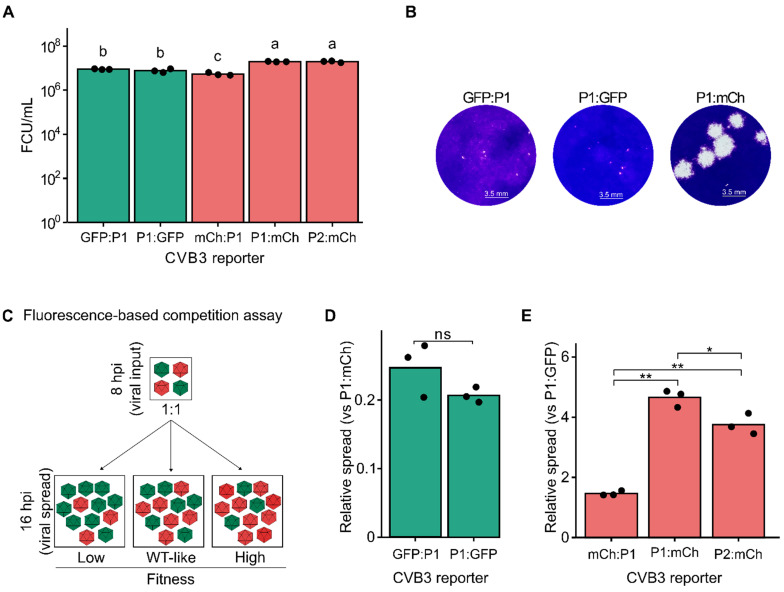
Analysis of CVB3 reporter virus fitness. (**A**) Single-cycle virus production. Viral titration in HeLa-H1 cells after infection at a multiplicity of infection (MOI) of 5. Bars represent the mean of three biological replicates, shown as black dots. Statistical significance was determined using one-way ANOVA followed by Tukey’s post hoc test (*p* < 0.05) on log-transformed data. Groups sharing the same letter are not significantly different. (**B**) Representative images of plaque morphology for CVB3 GFP:P1, P1:GFP, and P1:mCh reporter viruses. (**C**) Schematic representation of the fluorescence-based competition assay. mCherry- and eGFP-expressing reporter viruses (colored in red or green, respectively) are mixed at a 1:1 ratio and used to infect cells. Images are taken at 8 hpi and 16 hpi to measure the number of cells infected with each virus following initial infection and viral spread, respectively. Relative fitness can then be calculated by dividing the number of fluorescent cells at 16 hpi by those at 8 hpi for each reporter virus. (**D**,**E**) Relative viral spread (16 hpi versus 8 hpi) comparing GFP-encoding viruses (GFP:P1 or P1:GFP) against an mCherry-encoding control virus (P1:mCh; **D**), and mCherry-encoding viruses (mCh:P1, P1:mCh, or P2:mCh) against a GFP-encoding virus (P1:GFP; **E**). Statistical significance was assessed using unpaired *t*-tests (n = 3). ns, *p* > 0.05; *, *p* < 0.05; **, *p* < 0.01.

**Figure 3 viruses-17-00796-f003:**
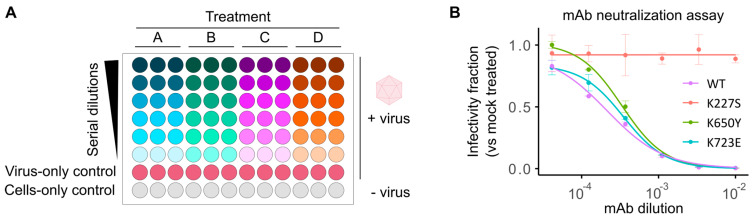
Fluorescent reporter viruses for screening neutralizing antibodies. (**A**) Schematic representation of the sera neutralization assay workflow. A fixed amount of virus is incubated with serial dilutions of sera or monoclonal antibodies (Treatments: A, B, C, D) before infecting cells. No-sera (virus-only) and no-virus (cells-only) controls are included to determine maximum infectivity and background fluorescence levels, respectively. The number of fluorescent cells or average well fluorescent intensity is quantified following a single cycle of infection (e.g., 8 hpi). Background fluorescence (cells-only control) is subtracted from all virus-infected cells. Subsequently, the relative signal in each infected well is then standardized to that of untreated, virus-infected controls (virus-only control) followed by curve-fitting to determine the concentration resulting in the desired inhibition level (e.g., IC50, IC90, etc.). (**B**) Neutralization of WT and mutant reporter viruses by a monoclonal antibody (mAb). Data represent mean ± SD from two independent experiments.

**Figure 4 viruses-17-00796-f004:**
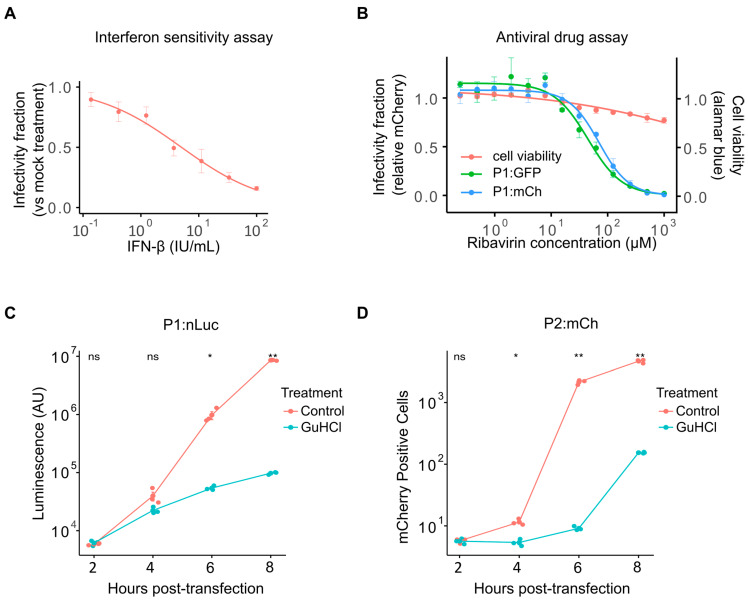
Antiviral screening and mechanism of action studies using fluorescent viruses. (**A**) Dose–response curve of interferon (IFN) sensitivity for the WT CVB3 fluorescent reporter virus. (**B**) Sensitivity of GFP- and mCherry-expressing viruses to ribavirin treatment determined at 16 h post-infection (hpi). Cell viability was assessed in parallel using the resazurin reduction (Alamar blue) assay (red line). (**C**,**D**) Growth curve for P1:nLuc (**C**) and P2:mCh (**D**) in HeLa-H1 cells. Cells were transfected with in vitro-transcribed RNA and either mock-treated or treated with 0.5 mM GuHCl, and reporter signal was quantified at 2 h intervals. The experiment was performed in triplicate and mean values ± SD are shown. Statistical significance was assessed using *t*-tests on log-transformed data. ns, *p* > 0.05; *, *p* < 0.05; **, *p* < 0.001.

**Figure 5 viruses-17-00796-f005:**
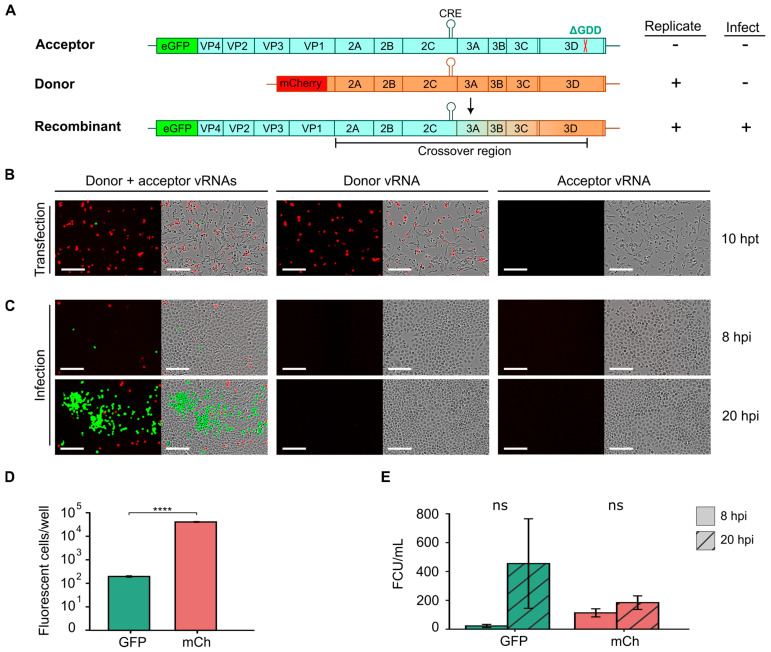
A quantitative, fluorescence-based recombination assay. (**A**) Schematic diagram of CVB3 acceptor and donor vRNAs, along with the recombinant vRNA product. The acceptor genome (blue) encodes eGFP but is rendered replication-incompetent by deletion of the catalytic triad in the viral polymerase (ΔGDD). The donor genome (orange) is a subgenomic replicon in which the capsid region has been replaced with mCherry. Following co-transfection, recombination can yield a replication-competent genome encoding the eGFP reporter. The color gradient (from blue to orange) between the 3′ end of the VP1-coding region and the ΔGDD indicates the crossover region within which recombination must occur to produce a functional genome. (**B**) Live-cell fluorescence and phase contrast images from the cell-based recombination assay. In vitro transcribed vRNAs were transfected into BHK-21 cells and reporter expression was examined at 10 hpt to quantify the number of eGFP- or mCherry-positive cells. (**C**) Supernatants collected at 10 hpt from (**B**) were used to infect HeLa-H1 cells and the fluorescence and phase contrast images at 8 and 20 hpi are shown. (**D**,**E**) Quantification of the number of eGFP- or mCherry-positive cells from the transfection experiments (**D**) or the infection experiment (**E**). Scale bar: 200 µm; CRE, *cis*-replication element. ns, *p* > 0.05; ****, *p* < 0.0001 by two-tail *t*-test on log-transformed data.

## Data Availability

All primary data and reagents are available upon request from RG. Scripts for antiviral and sera neutralization can be found at https://github.com/RGellerLab/EC-calculation-scripts, accessed on 14 April 2025.

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
