# Peer review of "A Fluorescent Reporter Virus Toolkit for Interrogating Enterovirus Biology and Host Interactions"

_viruses, 2025, doi:10.3390/v17060796_

Round 1
Reviewer 1 Report
Comments and Suggestions for Authors
This paper presents a versatile set of genetically engineered fluorescent Coxsackievirus B3 (CVB3) reporter viruses designed to facilitate real-time studies of enterovirus biology. The authors compare multiple insertion sites for fluorescent reporters (eGFP and mCherry) within the viral polyprotein and identify the most permissive regions for maintaining viral fitness. They develop and validate assays using these reporter viruses to measure viral fitness, perform antibody neutralization screens, assess antiviral drug efficacy, and analyze viral recombination. The authors also compare the replication kinetics of viruses encoding either luminescent or fluorescent reporters. The authors suggest that the fluorescent reporter viruses toolkit provides a fast, cost-effective, and scalable platform for studying viral dynamics, with broad applicability to other enteroviruses due to conserved genome features.
The authors provide no information on the coding sequence for the fluorescent reporters. Are the sequences optimized in a way to limit the number of CpG and UpA dinucleotides? Many RNA viruses are known to have sequences with these dinucleotide pairs underrepresented to prevent host pattern recognition receptor activation. Also, do the coding sequences match the frequency of codons found in CVB3? These could be important considerations when designing these reporter viruses to improve viral fitness and retain the encoded sequence. At least a comment on this in the manuscript might be appropriate.
Do the authors have a reason for the loss of the eGFP sequence from the P2 and P3 region? Was this repeated multiple times with different cDNA clones yielding the same result?
How stable are the fluorescent reporter viruses to subsequent viral passage(s)? Do the viruses lose the sequences after 2,3,4 or more passages?
Did the authors determine the genome:PFU ratio for the fluorescent reporters? How does this compare to CVB3 with no encoded fluorescent reporter sequence?
Do the authors observe similar dose response cure in Figure 4B if they use the P1:nLuc virus? Having the comparison in Figure 4C and 4D on the replication kinetics is set up to ask this question. How do the IC50 values for ribavirin agree with literature values?
Author Response
We would like to thank the reviewer for the insightful comments. Below is a point-by-point response to each of the points raised.
Comment 1: The authors provide no information on the coding sequence for the fluorescent reporters. Are the sequences optimized in a way to limit the number of CpG and UpA dinucleotides? Many RNA viruses are known to have sequences with these dinucleotide pairs underrepresented to prevent host pattern recognition receptor activation. Also, do the coding sequences match the frequency of codons found in CVB3? These could be important considerations when designing these reporter viruses to improve viral fitness and retain the encoded sequence. At least a comment on this in the manuscript might be appropriate.
Response 1: We apologize for not making this clear in the original text. The sequences of all plasmids used are available in the supplement. We have now indicated that the fluorescent reporter sequence was not altered in the methods section. As mentioned in the main text (line 154), we agree that differences between eGFP and mCherry reporters could be due to codon usage, among other reasons.
Comment 2: Do the authors have a reason for the loss of the eGFP sequence from the P2 and P3 region? Was this repeated multiple times with different cDNA clones yielding the same result?
Response 2: Yes, we have attempted the recovery of the P2:GFP virus multiple times from different bacterial clones and always failed to recover fluorescent virus. We have also revalidated the plasmid sequence by full plasmid sequencing to ensure the sequence was correct. Hence, we are confident that the inability to observe fluorescent signal following the initial replication is due to the strong fitness costs imparted by the insertion of the eGFP sequence at this position, resulting in a loss of the insert starting in the passage 0 virus stock.
Comment 3: How stable are the fluorescent reporter viruses to subsequent viral passage(s)? Do the viruses lose the sequences after 2,3,4 or more passages?
Response 3: We have not directly assayed this for our reporter viruses, although there are numerous examples in the literature showing the loss of insert upon serial passaging. As indicated in the manuscript, we only use early passage virus to avoid the loss of the reporter and all the experimental workflows described in this manuscript are amenable to the use of such early passage fluorescent reporter viruses.
Comment 4:Did the authors determine the genome:PFU ratio for the fluorescent reporters? How does this compare to CVB3 with no encoded fluorescent reporter sequence?
Response 4: We have not investigated this question as none of our experimental workflows relies on this information.
Reviewer 2 Report
Comments and Suggestions for Authors
This study designed and validated fluorescent reporter virus tool kit by inserting fluorescent proteins in different regions in enterovirus genomes using Coxsackievirus B3, which offers a platform for enterovirus research in different aspects, including analyzing viral replication, host interactions and antiviral responses and etc. This is obviously a useful tool. However, there are still things to be improved, and this study could be more enriched in its design.
- For references, please go over your manuscript and make sure that each solid statement has a reference. For instance, line 29-33, 65-71 miss references.
- The overall flow of the introduction sounds weird. You may want to break the second paragraph of the introduction into 2 paragraphs, so you don’t pile up different information at once. Also, please format your manuscript nicely. Some are single spaced, but some are double spaced. It seems like this manuscript is assembled by different people.
- You found a specific good inserting position P1/P2 and it may be applied to other enteroviruses such as EV71 and echoviruses. It would be great to have a sequence alignment/comparison figure/data here. It’s also nice to prove your findings in other important enteroviruses such as EV71.
- It would be nice to have some cryo-EM structural data to prove that fluorescent modification will not impact the capsid structure framework. Please at least include the structural and non-structural genetic information in the introduction.
Author Response
We would like to thank the reviewer for their helpful input. Below is a point-by-point response to each of the points raised.
Comment 1: For references, please go over your manuscript and make sure that each solid statement has a reference. For instance, line 29-33, 65-71 miss references.
Response 1: We thank the reviewer for pointing this out and have included two additional references to better support these statements.
Comment 2:The overall flow of the introduction sounds weird. You may want to break the second paragraph of the introduction into 2 paragraphs, so you don’t pile up different information at once. Also, please format your manuscript nicely. Some are single spaced, but some are double spaced. It seems like this manuscript is assembled by different people.
Response 2: We thank the reviewer for their suggestion and have split the paragraph into two sections.
Comment 3: You found a specific good inserting position P1/P2 and it may be applied to other enteroviruses such as EV71 and echoviruses. It would be great to have a sequence alignment/comparison figure/data here. It’s also nice to prove your findings in other important enteroviruses such as EV71.
Response 3: To address the reviewer’s comment, we have included a sentence about the successful insertion of reporter viruses at this position for different enteroviruses, including EV71. As this has been previously demonstrated to work in various enteroviruses, we do not believe it is necessary to reproduce this.
Comment 4: It would be nice to have some cryo-EM structural data to prove that fluorescent modification will not impact the capsid structure framework. Please at least include the structural and non-structural genetic information in the introduction.
Response 4: We believe this is outside the scope of the work as it does not directly relate to the experimental workflows described in the manuscript.